# Structural Characterization and Anticancer Activity of a New Anthraquinone from *Senna velutina* (Fabaceae)

**DOI:** 10.3390/ph16070951

**Published:** 2023-07-01

**Authors:** David Tsuyoshi Hiramatsu Castro, Daniel Ferreira Leite, Debora da Silva Baldivia, Helder Freitas dos Santos, Sikiru Olaitan Balogun, Denise Brentan da Silva, Carlos Alexandre Carollo, Kely de Picoli Souza, Edson Lucas dos Santos

**Affiliations:** 1Research Group on Biotechnology and Bioprospecting Applied to Metabolism (GEBBAM), Universidade Federal da Grande Dourados, Dourados 79804-970, Brazil; david_hiramatsu@hotmail.com (D.T.H.C.); danielleitesci@gmail.com (D.F.L.); helderspk@gmail.com (H.F.d.S.); sikirubalogun@ufgd.edu.br (S.O.B.); kelypicoli@ufgd.edu.br (K.d.P.S.); 2Programa de Pós-Graduação em Ciências da Saúde, Universidade Federal da Grande Dourados, Dourados 79804-970, Brazil; 3Laboratory of Natural Products and Mass Spectrometry, Universidade Federal do Mato Grosso do Sul, Cidade Universitária, Campo Grande 79070-900, Brazil; denisebrentan@gmail.com (D.B.d.S.); carlos.carollo@ufms.br (C.A.C.)

**Keywords:** melanoma, leukemia, medicinal plants, anthracene, caspase

## Abstract

In this study, a novel compound was isolated, identified, and its chemical structure was determined from the extract of the roots of *Senna velutina*. In addition, we sought to evaluate the anticancer potential of this molecule against melanoma and leukemic cell lines and identify the pathways of cell death involved. To this end, a novel anthraquinone was isolated from the barks of the roots of *S. velutina*, analyzed by HPLC-DAD, and its molecular structure was determined by nuclear magnetic resonance (NMR). Subsequently, their cytotoxic activity was evaluated by the (3-(4,5-dimethylthiazol-2-yl)-2,5-diphenyl-2H-tetrazolium bromide (MTT) method against non-cancerous, melanoma, and leukemic cells. The migration of melanoma cells was evaluated by the scratch assay. The apoptosis process, caspase-3 activation, analysis of mitochondrial membrane potential, and measurement of ROS were evaluated by flow cytometry technique. In addition, the pharmacological cell death inhibitors NEC-1, RIP-1, BAPTA, Z-VAD, and Z-DEVD were used to confirm the related cell death mechanisms. With the results, it was possible to elucidate the novel compound characterized as 2′-OH-Torosaol I. In normal cells, the compound showed no cytotoxicity in PBMC but reduced the cell viability of all melanoma and leukemic cell lines evaluated. 2′-OH-Torosaol I inhibited chemotaxis of B16F10-Nex2, SK-Mel-19, SK-Mel-28 and SK-Mel-103. The cytotoxicity of the compound was induced by apoptosis via the intrinsic pathway with reduced mitochondrial membrane potential, increased levels of reactive oxygen species, and activation of caspase-3. In addition, the inhibitors demonstrated the involvement of necroptosis and Ca^2+^ in the death process and confirmed caspase-dependent apoptosis death as one of the main programmed cell death pathways induced by 2′-OH-Torosaol I. Taken together, the data characterize the novel anthraquinone 2′-OH-Torosaol I, demonstrating its anticancer activity and potential application in cancer therapy.

## 1. Introduction

Cancer is the second leading cause of death worldwide, accounting for approximately 10 million deaths per year [1]. Leukemia is a malignant neoplasm of hematopoietic tissue characterized by abnormal proliferation of white blood cells as the progenitor cells become unable to differentiate and/or mature in the bone. This type of neoplasm can be classified as lymphocytic or myeloid according to the cellular origin, and as acute or chronic based on the disease progression [2].

Melanoma is the most aggressive skin cancer, given its high metastatic capacity and lethality [3]. This type of cancer affects melanocytes, cells that produce the pigment melanin [4]. Although melanoma cases account for only 3% of all skin cancer cases, it is associated with the highest mortality rates (about 65%) [5]. The most common treatments for leukemia are chemotherapeutic drugs, radiotherapy, and hematopoietic stem cell transplantation [6], while for melanoma there is surgical removal, radiotherapy, and chemotherapy, which are considered mostly for cancer identified in its early stages. Despite the low prognosis in advanced stages, mainly due to metastasis, the indicated treatment is immunotherapy, together with other chemotherapeutic drugs [7,8]. However, despite the positive results found with immunotherapy, this treatment is still not widely used in clinical practice because of its high cost [9].

Although chemotherapies are important tools for the treatment of different types of cancer, many of the currently used drugs have low selectivity, different levels of systemic toxicity, and cellular resistance to chemotherapeutic agents [10,11]. Thus, natural molecules arising from plants have played a significant role in the development of new chemotherapeutic agents by having a wide structural variety with antitumor potential [12,13]. In this way, plants can be investigated for the discovery of new anticancer drugs with greater selectivity, efficacy, and reduced toxicity, impacting positively on a good prognosis, longer life expectancy, and better quality of life for patients.

Following this premise, Campos et al. [14] have shown that the extract of the leaves of *Senna velutina*, a shrub belonging to the Fabaceae family, showed antileukemic properties. Later, the study by Castro et al. [15] demonstrated antitumor activity of the extract of the roots of *S. velutina*, against murine melanoma cells in in vitro and in vivo models. In this study, in addition to the mechanisms involved, the ability of the extract to reduce subcutaneous tumor volume and prevent lung metastasis progression in mice was also substantiated. Among the chemical constituents identified in the extract, flavonoid derivatives, catechins, piceatannol, and anthraquinones stand out as potentially responsible for the observed antitumor effects.

Among the compounds identified in *S. velutina*, the anthraquinones have been highlighted as a class of compounds with intriguing pharmacological potential. The number of studies demonstrating the potential efficacy of anthraquinones as antitumor agents has been on the rise [16]. Studies by Chiang et al. [17] have demonstrated the cytotoxic activity of anthraquinones via induction of DNA damage in tumor cells. Furthermore, anthraquinones in preclinical studies such as emodin and aloe-emodin have been shown to be able to induce apoptosis and cell cycle arrest [18,19]. Additionally, there is evidence that anthraquinones can promote inhibition of metastasis formation and induce different types of cell death [20,21,22].

Considering the anticancer potential of anthraquinones and continuing our previous research, this study aimed to isolate, identify, characterize, and determine the chemical structure of the major anthraquinone identified from the extract of *S. velutina*, roots. In addition, this study sought to evaluate its cytotoxic potential against melanoma and leukemic cell lines and identify the possible death pathways involved.

## 2. Results

### 2.1. Structural Determination of 2′-OH-Torosaol I, Which Is Also Referred to as 2,2′-OH- Singueanol I (***1***)

Compound **1** was obtained through column chromatography from the root barks of *S. velutina* chloroform extract. Based on LC-MS analysis, we found it to have a purity of 93%. Additionally, its UV spectrum shows absorption bands at 278, 321, 333, and 406 nm wavelengths (λmax), suggesting that it has a bis-tetrahydro-anthracene chromophore. The mass spectrum obtained by positive ion mode revealed an intense ion at *m*/*z* 635.2128 [M+H]^+^, confirming the molecular formula C_34_H_34_O_12_ (error 0.8 ppm). From MS/MS of ion *m*/*z* 635.2128, the product ions yielded by consecutive losses of H_2_O and CO were observed.

From the ^1^H NMR spectra, as demonstrated in Table 1, it was possible to confirm the symmetry for **1**. An aromatic proton was observed at δ 5.96 (*s*), which revealed heteronuclear correlations with the carbons at δ 8.1 (C-7/7`-Me), 122.9 (C-10/10′), 109.7 (C-7/7′), 122.9 (C-10/10′), and that it was compatible to H-5/5′. A methoxy group was confirmed by signals δ 3.38 (*s*) and 55.1, which revealed correlations in the HMBC spectra with the carbon at δ 161.2 (C-6/6′). An oxymethine proton was observed at δ 4.35 (*s*), which was attributed to position 2/2′ and correlated to carbons δ 205.2 (C-1/1′), 73.1 (C-3/3′), and 25.9 (C-3/3′-Me). Additionally, two methyl groups were observed with chemical shifts of δ 1.09 (*s,* 6H) and 2.01 (*s,* 6H), and only one methylene carbon was visualized in the DEPT 135° experiment at δ 39.6 (C-4/4′) that correlated to hydrogens δ 2.32 (*d, J* = 16.8 Hz) and 2.71 (*d, J* = 16.8 Hz) in the HSQC. Some chemical shifts were compared with the reported for similar substances [23,24]. Thus, the substance was structurally characterized as the new 10,10′-coupled dihydro-anthracene 2′-OH-Torosaol I (Figure 1), which is also referred to as 2,2′-OH-singueanol I.

### 2.2. Effect of 2′-OH-Torosaol (***1***) on Cell Viability

Effect of 2′-OH-Torosaol I on cell viability in non-cancerous and cancerous cell lines. PBMC (non-cancer), melanoma (B16F10-Nex2, SK-Mel-19, SK-Mel-28, SK-Mel-103), and leukemic (Jurkat and K562) cells were incubated at different concentrations of 2′-OH-Torosaol I for 24 h and 48 h, and viability was assessed by the MTT method. At the concentrations evaluated, 2′-OH-Torosaol I maintained the viability of PBMCs of 70% at the highest concentration evaluated, and the IC_50_ therefore is above 630 µM/mL (Table 2 and Figure 2A). 2′-OH-Torosaol I displays selectivity for cancer cells and in Figure 2B–G it can be seen that it significantly reduced the viability of all cell lines evaluated in a dose- and time-dependent manner while sparing the normal cell (PBMC).

Table 2 presents the respective IC_50_ values for each cell line evaluated. Comparing the IC_50_ between 24 h and 48 h of the cell lines tested, all demonstrated lower IC_50_ at 48 h except SK-Mel-103.

The results were obtained by 3-(4,5-dimethylthiazol-z-yl)-2,5-diphenyltetrazolium bromide (MTT) assay after 24 h and 48 h of incubation. Data are presented as mean ± SEM of three separate experiments, with three replicates in each experiment, and where IC_50_ represents half-maximal inhibitory concentration.

### 2.3. Evaluation of 2′-OH-Torosaol (***1***) in Suppressing Cell Migration of Melanoma Cells

Melanoma cells are highly metastatic, both in vitro and in vivo. After observing the cytotoxic effect on the viability of the tested cells, described in Table 1, the effect of 2′-OH-Torosaol I on cell migration was performed at a concentration of ⅛ of the IC_50_ value of the respective cell lines and determined by scratch assay. In Figure 3A–D, it is possible to observe, through the representative images, that the compound reduced the migration of all melanoma cells evaluated. In Figure 4A, we demonstrate that 2′-OH-Torosaol I at the concentration of 0.725 µM (equivalent to 1/8 of the IC_50_) was able to significantly inhibit the migration of melanoma cells in all tested cell lines. The greatest inhibitory effect of 2′-OH-Torosaol I was observed by inhibiting cell migration by 50% for B16F10-Nex2 and by 55% for SK-Mel-19, when compared with the control, at the final time of 48 h (Figure 3A,B).

### 2.4. Analysis of the Cell Death Profile Induced by 2′-OH-Torosaol I

Annexin V-FITC/PI labeling was used to assess the involvement of cell death by apoptosis. As shown in Figure 5B, 2′-OH-Torosaol I induced late apoptosis by double-labeling the fluorophores in the Jurkat leukemic lineage (58% versus 11% of control cells). For the K562 leukemic lineage (Figure 5D), there was increased double labeling, also indicating late apoptosis (32% versus 11% of control cells). For the B16F10-Nex2 melanoma lineage (Figure 5E,F), there was increased double labeling, also indicating late apoptosis (34% versus 7% of control cells).

### 2.5. Assessment of Mitochondrial Membrane Potential (MMP) in Melanoma and Leukemic Cells

The results obtained by flow cytometry using a JC–1 marker demonstrate that 2′-OH-Torosaol I caused impairment in mitochondrial function. As observed in the results, when compared with untreated cells, 2′-OH-Torosaol I decreased mitochondrial potential, demonstrated by altered red fluorescence and increased green fluorescence. The depolarization percentage was reduced by approximately 89%, 90%, and 91% for Jurkat, K562, and B16F10-Nex 2 strains, respectively (Figure 6A,C,E).

### 2.6. Determination of Levels of Reactive Oxygen Species (ROS)

By evaluating the levels of reactive oxygen species in the intracellular medium using the fluorescence marker 2′,7′-dichlorofluorescein diacetate (DCFH-DA), it was observed that 2′-OH-Torosaol I significantly increased the levels of ROS when compared with the control group in all three cancer cell lines evaluated (Figure 7A,C,E).

### 2.7. Evaluation of Activated Caspase-3 Protein

After verifying the reduction in mitochondrial potential, the involvement of activated caspase-3 was investigated. Quantification of activated caspase-3 was undertaken by labeling with anti-caspase-3 conjugated to a fluorophore and detected by flow cytometer. Figure 8 shows that incubation of cancer cell lines with 2′-OH-Torosaol I significantly increased activated caspase-3 in all three cell lines evaluated when compared with the control group. The greatest involvement occurred in K562 cells (Figure 8C,D), where the increase in activated caspase-3 was approximately 3.5-fold higher than in the control group.

### 2.8. Cell Death Analysis Using Inhibitors of Cell Death Pathways

To confirm cell death pathways, we used pharmacological inhibitors of the related pathways during the MTT cell viability assay. As shown in Figure 9, the cytotoxicity promoted by 2′-OH-Torosaol I after 24 h is related to increased caspase activity when pre-incubated with Z-DEVD and Z-VAD, pathways that are involved in apoptosis. Other pathways were also observed when pre-incubated with a necrostatin-1 inhibitor, which is related to necroptotic cell death. In addition, there was an increase in intracellular calcium concentration in the treatment pre-incubated with calcium chelator BAPTA. Reactive species are also related, as there was inhibition of cell death when pre-incubated with NAC antioxidant.

## 3. Discussion

When considering the vast world biodiversity and its potential for the development of new drugs of natural origin, the number of existing studies is still limited. Moreover, each plant species presents several biosynthetic pathways capable of producing molecules with great structural diversity and potential for different pharmacological applications, so much so that previous studies with molecules of plant origin have made possible the development of drugs such as vincristine, paclitaxel, irinotecan, and roscovitine, used worldwide as anticancer drugs. [25]. In this perspective, plants can be seen as a promising source of new, as of yet unidentified bioactive compounds that are considered strong candidates to expand the current list of antineoplastic agents [26]. Among the molecules produced by plants, there is the class of anthraquinones, an important group of secondary metabolites with a wide variety of biological applications and more than 700 chemical structures already identified [16,27]. Its biological activities include laxative [28], antifungal [29], antibacterial [30], antiviral [31], anti-inflammatory [32], and anticancer effects [33,34].

The class of anthraquinones is playing a significant role in the discovery of new antitumor agents, considering that several chemotherapeutic agents currently approved for clinical use in cancer therapy have anthraquinones in their molecular structure, for example, mitoxantrone, doxorubicin, epirubicin and dalrrubicin [21]. However, there is an emerging limitation related to the poor efficacy of some chemotherapeutic agents, due to the resistance of tumor cells to the drugs, as well as to the damage caused in the body due to toxic and side effects [35,36]. Considering the mentioned difficulties, the search for new anticancer molecules, such as new anthraquinones and their derivatives, more selective and with fewer side effects will allow the development of new anticancer drugs.

In previous studies, Campos et al. [14] demonstrated the anticancer activity of the crude extract of *Senna velutina* leaves on Jurkat and K562 leukemia cell lines. In that study, the extract showed apoptotic cytotoxicity by increasing caspase-3 activity and decreasing mitochondrial membrane potential. Subsequently, Castro et al. [15] demonstrated the anticancer activity of the crude extract from the roots on melanoma lineage in the in vitro and in vivo assays. The extract promoted death by apoptosis with the involvement of caspase-3, the same pathway promoted by the compound isolated in this study. Among the phytochemical constituents identified in the crude extract were anthraquinones, well described in the literature as being present in the chemical composition of the genus Senna [37,38]. Continuing from previous studies, in the present study, we isolated, identified, and characterized the structure of a previously undescribed bi-tetrahydro-anthracene from the barks of S. *velutina* root. The structural determination of the compound was elucidated by magnetic resonance imaging as 2′-OH-Torosaol I, which is compatible with the bis-tetrahydro-anthracene chromophore, similar to those isolated from the roots of *Senna singueana.* In the literature, there are similar compounds, differing in the position of one hydroxyl (OH) in their molecular structure, found in the species *Cassia torosa*, which has also been described for its anticancer activity [39].

2′-OH-Torosaol I demonstrated marked cytotoxic activity against different cancer cell lines. The cancer cells most sensitive to cytotoxic activity were B16F10-Nex2 and leukemic Jurkat and K562. Compared with studies conducted with the crude extract of *S. velutina*, the isolated compound was almost twenty times more potent concerning IC_50_. On the other hand, when evaluated against peripheral blood mononuclear cells (PBMC), 2′-OH-Torosaol I showed no relevant toxicity, indicating selectivity against cancer cells. This selectivity is a significant finding because the major limitations of chemotherapy are related to systemic toxicity and damage caused to normal cells in the body by chemotherapeutic agents, which can compromise and impede the progression of treatment [40,41].

Besides the cytotoxic effect, our study also demonstrated an inhibitory effect on tumor cell migration. This inhibition plays a relevant role, mainly, in the initial formation and progression of the tumor. Moreover, cell migration also contributes to the metastatic development of tumor cells. Thus, the inhibition of cell migration, being essential in the stages of metastasis formation, becomes an interesting therapeutic strategy for the development of antitumor drugs from natural molecules [22,42,43,44]. Thus, we investigated the effect of 2′-OH-Torosaol I, at a sub-lethal concentration, on the migration of melanoma cells in a medium without exogenous growth factors and the results strongly suggest that there was strong inhibition of migration in all melanoma cell lines evaluated.

Regarding the mechanism of action associated with cytotoxicity in Jurkat and K562 cell lines, our studies demonstrated the action of 2′-OH-Torosaol I on apoptosis-related death pathways. On analyzing the cell death profile, a predominance of late apoptosis was noted [45]. This type of apoptosis presents mechanisms similar to those of necrosis and does not involve phagocytic processes of apoptotic bodies. Late apoptosis exhibits all the morphological characteristics of apoptosis and activation of the caspase cascade, but also other events characteristic of necrosis, such as cytoplasmic swelling and loss of membrane integrity [45,46,47]. One of the features of apoptosis is the externalization of phosphatidylserine, a marker of Annexin V-FITC, which occurs due to the activation of proteases that translocate this phospholipid to the outside [48,49]. Anthraquinones and their derivatives are reported in the literature to promote cell death by apoptosis in tumor cell lines [21]. Anthraquinone-derived compounds, such as hypericin, also promote cytotoxicity through the same signaling pathway [50]. Other examples, such as Senoside A, with a similar molecular structure to 2′-OH-Torosaol I, are described to present cytotoxic activity with apoptotic death mechanism [51,52].

Functionally, mitochondrial activity is linked to cell death cascade by apoptosis [53]. Among the strategies adopted in anticancer therapy is the ability of some drugs to promote dysfunction in mitochondrial membrane potential (Δψm) [54]. The decrease in membrane potential is associated with the opening of pores in the membrane and permeabilization of mitochondria, resulting in the release of various apoptotic factors such as Cytochrome C, Smac, and Endo G into the cytoplasm, leading to apoptosis [55]. Our study revealed that 2′-OH-Torosaol I modified membrane functionality by decreasing membrane potential, indicating that 2′-OH-Torosaol I caused damage to both Jurkat and K562 leukemia cell lines mitochondria. This reduction may be related to Bcl-2 family proteins, central regulators of apoptosis [56]. This family is composed of antiapoptotic proteins such as Bcl-W, Bcl-XL, and Bcl-2 or proapoptotic proteins such as Bad, Bax, and Bid. The ratio of Bax/Bcl-2 determines the opening of pores in the mitochondrial membrane, initiating the apoptosis cascade [57]. Chemical structures similar to 2′-OH-Torosaol I induce apoptosis in tumor cells, evidenced by increased proapoptotic proteins and confirmed by reduced mitochondrial potential [51].

Mitochondrial membrane potential dysfunction recruits caspase family elements such as caspase-9 for apoptosome formation and, in sequence, the activation of caspase-3 [58]. Members of the protease family of caspases play an important role in the initiation and execution of apoptosis. Once activated, they cause proteolytic events, responsible for the classic features of apoptosis, which include nuclear condensation, DNA fragmentation, and formation of apoptotic bodies [59].

To confirm the mechanisms of death caused by the compound, inhibitors of cell death pathways were used. The data suggest that the main cause of death of the compound is related to the intrinsic apoptosis pathway. The cytotoxic activity of the compound indicates cell death by caspase-dependent apoptosis because there was an increase in cell viability when caspase inhibitors Z-VAD and caspase-3 inhibitors Z-DEVD were used, corroborating the data obtained in flow cytometry. In addition, the intrinsic pathway of apoptosis has, as a main factor, the dysfunction of mitochondrial membrane potential and the release of apoptotic factors [58]. The apoptosis pathways can be initially activated by increased reactive oxygen species (ROS) [60]. Given this, there was a partial increase in cell viability when incubated with the antioxidant NAC, indicating that there is involvement of the compound in the formation of ROS during the apoptosis cascade. Additionally, the data also suggest the involvement of intracellular Ca^2+^ after reducing cytotoxic activity in the presence of the inhibitor BAPTA. Increased Ca^2+^ in the cytoplasm can activate endonucleases, activate pro-apoptotic proteins, and initiate the death process [61,62]. However, activation of the necroptosis death pathway, a type of programmed necrosis death, has also been observed when incubating tumor cells with RIPK-1 inhibitor, NEC-1. RIPK-1 is a protein kinase responsible for initiating the necroptosis cascade [63].

Reactive oxygen species in tumor cells, at certain levels, can intensify oncogenic cascades such as proliferation, migration, and metastasis [64,65]. However, the excessive increase in ROS may be associated with cell death mechanisms such as apoptosis and other types [64]. Considering these facts, we demonstrated that 2′-OH-Torosaol I modulates the levels of ERO generation in melanoma and leukemic cells. The results demonstrate that anthraquinone 2′-OH-Torosaol I can promote an oxidative response, increasing the levels of ROS and contributing to cell death of cancer cells. This finding points to a potential therapeutic strategy against cancer because the increase in intracellular ROS in tumor cells intensifies signaling death pathways such as caspase-3 activation [64,66].

## 4. Materials and Methods

### 4.1. Collection and Preparation of Plant Material

The roots of *Senna velutina* were collected in November 2018, in the midwest region of Brazil (coordinates S 22°05′545″, W 55°20′746″) and identified by a botanist from the Universidade Federal da Grande Dourados (UFGD). For this purpose, the morphological and habitat parameters of the plant were observed. The collections were authorized by the Biodiversity Authorization and Information System (SISBIO) through protocol number 54470-1. The collected samples were deposited as specimens in the herbarium of UFGD, Dourados, Mato Grosso do Sul (MS), Brazil, under registration number 4665. After collection, the roots were washed and sanitized. The root peels were removed and placed in an oven with air circulation at 36 °C for 7 days and were then ground in a knife mill.

### 4.2. The Roots Extract Fractionating Process

*S. velutina* roots (395 g) were extracted by exhaustive percolation ethanol-water (7:3), for 48 h, dripping 45 drops per minute to obtain the ethanol extract. The collected material was rotary evaporated at 40 °C. The extract was submitted to liquid–liquid extraction with chloroform to obtain the chloroform extract (15.5 g). This extract was fractionated in classic silica gel column chromatography (Mesh 5 to 8) in elution gradient with chloroform and hexane 6:4 (*v*/*v*), chloroform and hexane 8:2 (*v*/*v*), chloroform, chloroform, and methanol 1%, chloroform and methanol 3%, chloroform and methanol 10% and methanol. In all, 30 fractions were obtained. Based on a thorough analysis using HPLC-DAD-MS, it became clear that the fractions 23–26 had almost identical chemical profiles. As a result, these fractions were combined for further examination., obtaining 450 mg that was reanalyzed by HPLC-DAD-MS and structurally characterized by NMR as 2′-OH-Torosaol I, which is also referred to as 2,2′-OH-singueanol I (93% purity grade), a novel substance in the literature.

### 4.3. HPLC-DAD-MS Analysis

The chemical analysis of the isolated compound was performed on a high-efficiency liquid chromatograph (Shimadzu, Kyoto, Japan) coupled to a diode array detector (DAD) and a MicrOTOF-Q III mass spectrometer (Bruker Daltonics, Billerica, USA) consisting of an electrospray ionization source (ESI). The capillary voltage applied for the mass spectrometry analyses was 4.5–3.5 kV, using nitrogen gas as a mist (4 bar), drying (9.0 L/min), and collision gas for the acquisition of the MS/MS analyses. The analysis was performed in negative and positive ionization modes (*m*/*z* 120–1200). The chromatographic column used was a Kinetex C-18 (2.6 µm, 150 × 2.2 mm, Phenomenex). The mobile phase used was composed of deionized water (solvent A) and acetonitrile (solvent B), both containing 0.1% formic acid (*v*/*v*). The flow rate was 0.3 mL/min, the injection volume was 1 µL, and during the analyses, the column was kept at 50 °C. The sample was prepared at a concentration of 1 mg/mL (methanol and deionized water, 6:4 *v*/*v*) and filtered through a Millex syringe filter (PTFE, 0.22 µm × 3.0 mm, Millipore) before injection into the chromatographic system. The elution gradient profile was the following: 0–2 min—3% of B, 2–25 min—3 to 25% of B, 25–40 min—25 to 80% of B, and 40–43 min 80% of B.

### 4.4. Structural Classification of theCompound by NMR

The compound was analyzed by magnetic resonance imaging (NMR; 1H, 13C, DEPT 135°, COSY, HMQC, and HMBC) on a Bruker DRX500 spectrometer (1H at 500 MHz and 13C at 125 MHz). Chemical shift (δ) was expressed in ppm, with relative values for TMS. The sample was solubilized in DMSO-D6.

### 4.5. Cell Culture

The cell lines used were human peripheral blood mononuclear cells (PBMC); Sk-Mel-19, Sk-Mel-28, Sk-Mel-103 human melanoma cells; B16F10-Nex2 murine melanoma cells; and K562 and Jurkat leukemic cells. For PBMC, B16F10-Nex2, K562, and Jurkat cells, RPMI medium was used. SK-Mel cell lines were cultured in DMEM high glucose medium. All (except PBMC) were kept in bottles containing their respective culture media and supplemented with 10% fetal bovine serum for the other strains, added 1% antibiotic (penicillin/streptomycin) at 10 mg/mL, conditioned in an incubator at 37 °C and 5% CO_2_.

For the preparation of human peripheral blood mononuclear cells (PBMC), human peripheral blood from a donor was collected in tubes containing sodium citrate anticoagulant. PBMC were isolated with Ficoll Histopaque-1077 (1.077 g/cm^3^) (Sigma Aldrich) according to the manufacturer’s instructions. The use of human blood was approved by the Ethics Committee of the Universidade Federal da Grande Dourados (UFGD) under protocol number 123/12.

### 4.6. Cell Viability Assay

To evaluate the cytotoxic effect of the compound on the viability of non-cancerous, melanoma, and leukemic cells, a colorimetric assay with 3-(4,5-dimethylthiazol-2-yl)-2,5-diphenyltrazolium bromide (MTT) was performed. For this, cells were plated at a density of 3 × 10^3^ for B16F10-Nex2, 7 × 10^3^ for Sk-Mel, 2 × 10^4^ for leukemic cells, and 12 × 10^4^ for PBMC in 96-well plates and treated with 2′-OH-Torosaol I (diluted in DMSO) at varying concentrations (2.46–630 µM). After 24 h and 48 h, the supernatant was discarded, washed with 1× PBS and 100 µL MTT (0.5 mg/mL) was added to each well. Upon incubation for 4 h in the incubator, the supernatant was removed and 100 µL of DMSO was added. Three independent experiments in triplicate were carried out (the reading was taken at 570 nm using a TP Reader (Thermo Plate)). Cell viability was calculated using the formula below:Cell viability (%) = [Abs treated cells/Abs control] × 100

In all cases, the control refers to untreated cells representing 100% viability.

### 4.7. Scratch Assay

To evaluate the effect of the compound on melanoma cell migration, the assay was performed according to Justus et al. [67], with minor modifications. For this purpose, the cells Sk-Mel-19, Sk-Mel-28, Sk-Mel-103 (5 × 10^4^) and B16F10-Nex2 (4 × 10^4^) were plated in 24-well plates and incubated in a CO_2_ oven at 37 °C. After reaching 90% confluence, a vertical scratch was made in the middle of each well with a 200 µL sterile tip. Then, each well was washed with 500 µL of 1× PBS to remove detached cells, and subsequently, the cells were incubated in a culture medium containing 1% SFB with 2′-OH-Torosaol I, diluted in 0.2% DMSO. Each well was photographed in three distinct regions with an inverted phase-contrast microscope (Olympus CKX41; 40× magnification) at the times of 0 h, 12 h, 24 h, and 48 h. To calculate the migration rate, the areas were determined at each time using ImageJ Software (NIII, Bethesda, MD, USA). Three independent experiments were performed in triplicates. For migration calculation, the following formula was used:“Gap closure “ (“%”)”= “(“Average time 0-Average of time evaluated”/“Average of time 0”)” × 100”

### 4.8. Flow Cytometry Evaluation of Apoptosis

The apoptotic effect of 2′-OH-Torosaol I treatment on Jurkat and K562 cells was evaluated, using the double labeling technique of Annexin V-FITC and Propidium Iodide (PI). Briefly, 10 × 10^4^ cells were plated in 12-well plates and treated with IC_50_ of 2′-OH-Torosaol I for 24 h and 48 h in a humidified atmosphere at 37 °C and 5% CO_2_. After each period, the cells were washed with binding buffer, resuspended, and labeled with Annexin V- Fluorescein Isothiocyanate (FITC) and PI according to the manufacturer’s instructions. Immediately thereafter, the cells were incubated for 30 min in a dark place. Fluorescence in the cells was detected by BD Accuri C6 Plus Flow Cytometer (BD Bioscience, San Diego, CA, USA) and analyzed in FlowJo 10.6 (flow cytometer software for analyzing images) for a total of 10,000 events. The population of Annexin-/IP- cells was considered normal, while the population of Annexin+/IP- was considered an indicator of apoptosis, the population of Annexin-/IP+ cells was considered an indicator of necrosis and finally, the Annexin+/IP+ cells population was considered an indicator of late apoptosis.

### 4.9. Measurement of Mitochondrial Membrane Potential (Δψm)

To evaluate MMP, labeling of cancer cells with 5,5′,6,6′-tetrachloro-1,1′, and 3,3′-tetraethyl-imidacarbocyanine iodide (JC–1) was performed according to the manufacturer’s instructions. Briefly, 10 × 10^4^ cells were plated in 12-well plates and treated with IC_50_ of 2′-OH-Torosaol I for 24 h in a humidified atmosphere at 37 °C and 5% CO_2_. After each period, cells were washed with 1× PBS, and 2 µM of JC–1 (Sigma Aldrich) was added to each sample and incubated at 37 °C for 30 min. The cells were washed again with 1× PBS and resuspended. To evaluate MMP, 10,000 events were detected on the BD Accuri C6 Plus Flow Cytometer (BD Bioscience, San Diego, CA, USA) and evaluated in FlowJo 10.6.

### 4.10. Measurement of the Generation of Intracellular Reactive Oxygen Species (ROS)

The production of ROS was measured using 2′,7′-dichlorofluorescein diacetate (DCFH-DA). Briefly, 10 × 10^4^ cells were plated in 12-well plates and treated with IC_50_ of 2′-OH-Torosaol I for 24 h. After each period, the cells were washed with 1× PBS, added to 10 µM DCFH-DA (Thermo Fisher), and incubated for 30 min in a dark place. To determine the amount of ROS produced, cells were washed with 1x PBS, resuspended, read on a BD Accuri C6 Plus Flow Cytometer (BD Bioscience, San Diego, CA, USA) and evaluated in FlowJo 10.6. A total of 10,000 events were collected.

### 4.11. Measurement of Activated Caspase-3

The involvement of caspase-3 in the cell death process was investigated with the Active Caspase-3 Apoptosis kit (BD Pharmigen) following the manufacturer’s instructions. Briefly, 10 × 10^4^ cells were plated in 12-well plates and treated with IC_50_ of 2′-OH-Torosaol I for 24 h. After the established period, the cells were washed with 1× PBS, centrifuged at 2000 rpm for 5 min at 10 °C and the supernatant was removed. The precipitate was washed twice more with PBS 1×. For each well, 2 µL of anti-caspase-3 was added and incubated for 1 h in the incubator. The reading was carried out in a BD Accuri C6 Plus Flow Cytometer (BD Bioscience, San Diego, CA, USA) and evaluated in FlowJo 10.6. A total of 10,000 events were collected [1].

### 4.12. Cellular Cytotoxicity Assay with Pharmacological Inhibitors

Jurkat, K562, and B16F10-Nex2 cells were plated in 96-well plates containing RPMI 1640 supplemented with 10% SFB in the presence of 20 µM pan-caspase inhibitor Z-VAD-FMK (N-Benzyloxycarbonyl-Val-Ala-Asp(O-Me) fluoromethyl ketone), 20 µM Z-DEVD, 20 µM of inhibitor Necrostatin-1 (NEC-1), 5 µM of inhibitor BAPTA and 50 µM of the antioxidant N-acetyl-cysteine (NAC) and incubated for 60 min in a humidified atmosphere at 37 °C and 5% CO_2_. Immediately after, the IC_50_ of 2′-OH-Torosaol I was added to each sample and incubated for 24 h. The cells were then incubated with 100 µL MTT (0.5 mg/mL) in each well. After 4 h in the oven at 37 °C, the supernatant was removed and 100 µL of DMSO was added. The reading was performed at 570 nm using TP Reader (Thermo Plate). The formula from Section 2.6 was used to calculate cell viability.

### 4.13. Statistical Analysis

All experiments were expressed as mean ± standard error of the mean (SEM) and performed at least three times. Results were analyzed using GraphPad Prism software (version 5.03; GraphPad Software, Inc., La Jolla, CA, USA). Differences between groups were evaluated by ANOVA analysis of variance, followed by the Tukey post hoc test. Data were considered significant when *p* > 0.05.

## 5. Conclusions

This study identified and characterized a novel anthraquinone in the barks of the roots of *Senna velutina*, with anticancer potential against melanoma and leukemia cells. The anthraquinone 2′-OH-Torosaol I showed increased selectivity for cancer cells and promoted cell death via apoptosis by decreasing mitochondrial membrane potential via increased caspase-3. In addition, it was able to inhibit the migration of tumor cells. Taken together, the results point to anthraquinone 2′-OH-Torosaol I as a molecule with potential application in therapeutics against diseases associated with cancer cell proliferation. As part of our future studies, we will explore further the mechanisms involved in anti-tumoral activities, including additional preclinical in vitro and in vivo studies of this novel anthraquinone.

## Figures and Tables

**Figure 1 pharmaceuticals-16-00951-f001:**
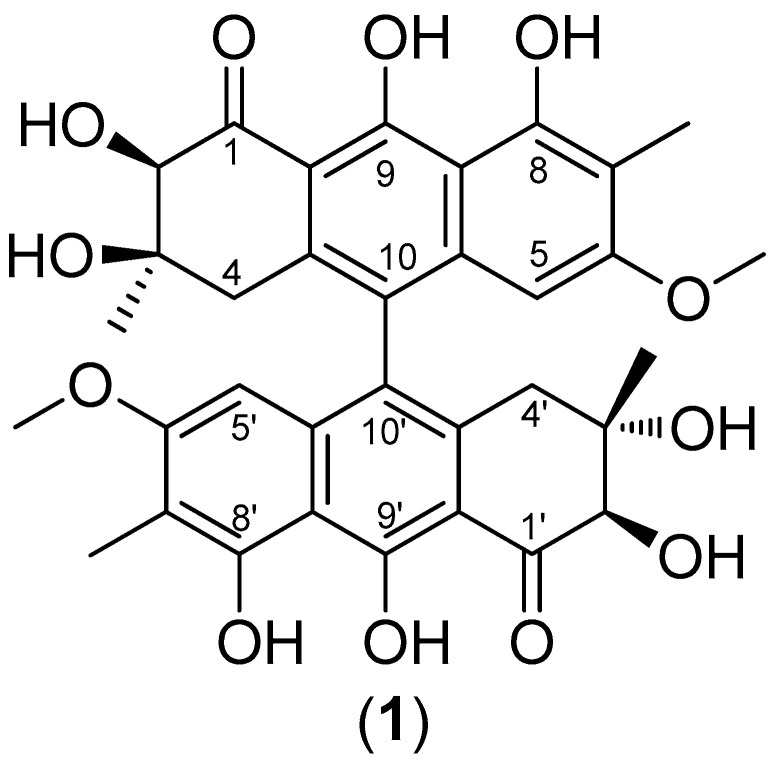
Structure of 2′-OH-Torosaol I.

**Figure 2 pharmaceuticals-16-00951-f002:**
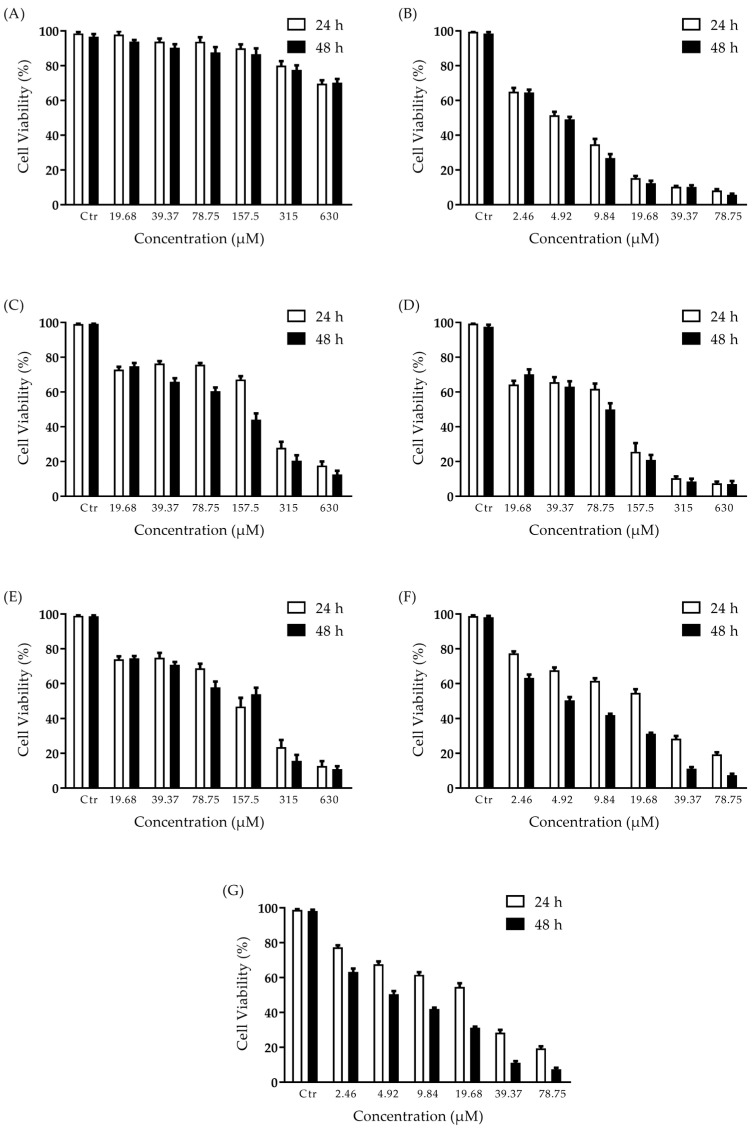
Cell viability determined by the MTT method. Cytotoxic effect of 2′-OH-Torosaol I against normal cell lines. (**A**) PBMC; melanoma cancer cells (**B**) B16F10-Nex2, (**C**) SK-Mel-19, (**D**) SK-Mel-28 (**E**) SK-Mel-103; and leukemic cancer cells (**F**) Jurkat and (**G**) K562, at 24 h and 48 h time points. Results are expressed as mean ± standard error of the mean (n = 3).

**Figure 3 pharmaceuticals-16-00951-f003:**
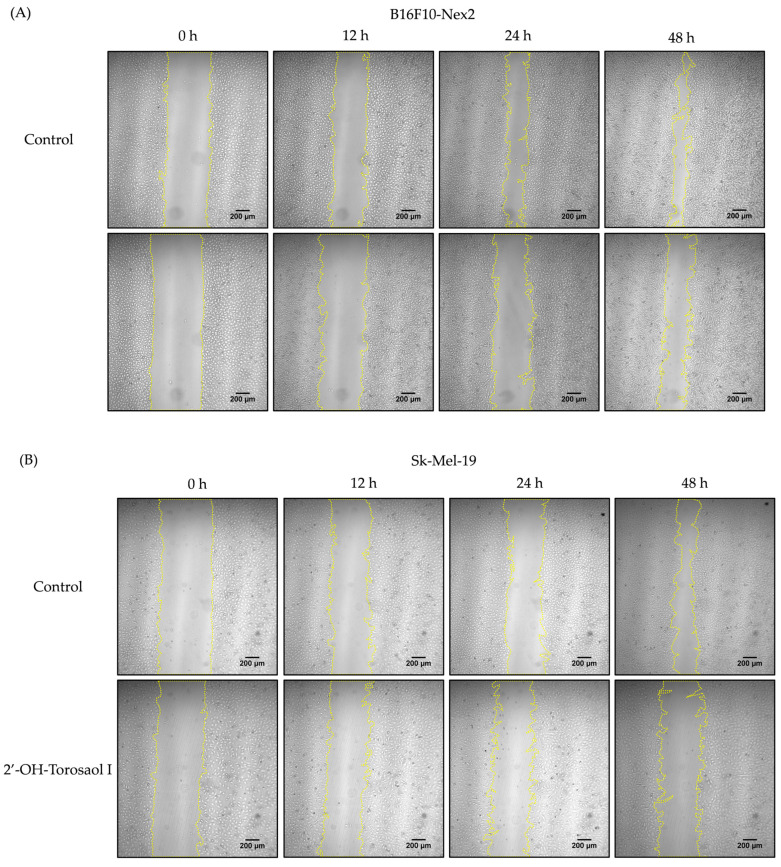
Representative images of melanoma cell migration at 0 h, 12 h, 24 h, and 48 h in the wound closing assay by scratch in 24-well plates treated with 0.725 µM concentration of 2′-OH-Torosaol I. The residual area of the groove was measured by Image J in melanoma cells (**A**) B16F10-Nex2, (**B**) SK-Mel-19, (**C**) SK-Mel-28 and (**D**) SK-Mel-103. Images were obtained by light microscopy at 40× objective. Scale bar 200 µm.

**Figure 4 pharmaceuticals-16-00951-f004:**
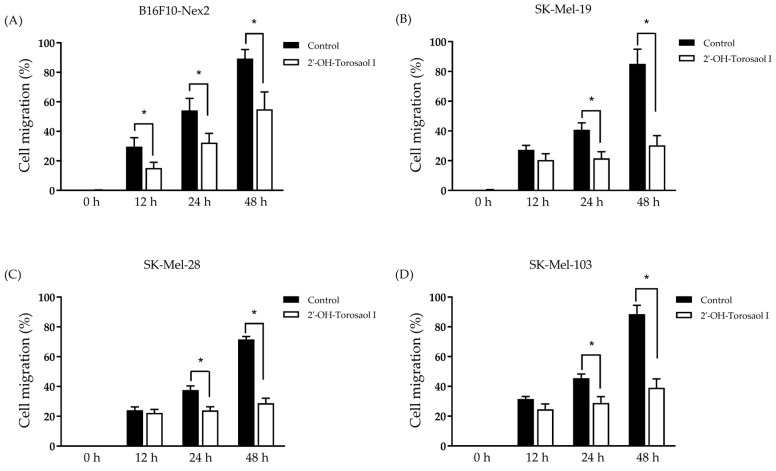
Cell migration analysis in the scratch assay wound closure after 0 h, 12 h, 24 h, and 48 h with 2′-OH-Torosaol I in (**A**) B16F10-Nex2 melanoma cells, (**B**) SK-Mel-19, (**C**) SK-Mel-28, and (**D**) SK-Mel-103. Results were obtained via three independent experiments and are expressed as mean ± standard error of the mean (n = 3) * *p* < 0.05 compared with the untreated control group.

**Figure 5 pharmaceuticals-16-00951-f005:**
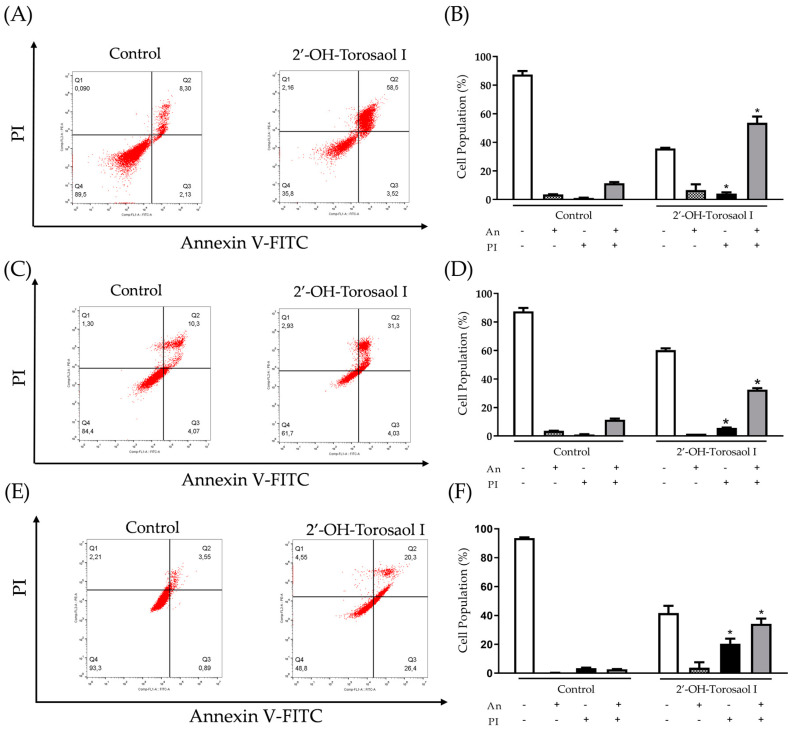
Induction of apoptosis in Jurkat and K562 leukemic cell lines and B16F10-Nex2 melanoma cell line. Representative dot plot of the cell population of (**A**) Jurkat, (**C**) K562, and (**E**) B16F10-Nex2. Representative graph showing the cell population at each type of labeling in the Annexin V-FIT/PI assay in (**B**) Jurkat, (**D**) K562, and (**F**) B16F10-Nex2 cells incubated with IC_50_ of 2′-OH-Torosaol I. Results are expressed as mean ± standard error of the mean (n = 3) * *p* < 0.05 when compared with the control group.

**Figure 6 pharmaceuticals-16-00951-f006:**
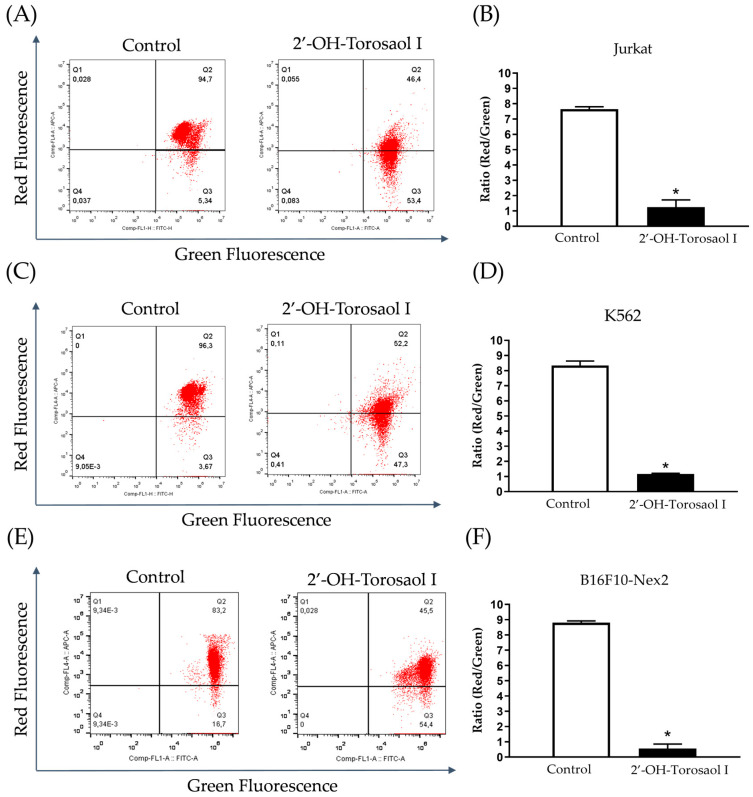
Dot plot of JC–1 analysis by flow cytometry in leukemia cell lines (**A**) Jurkat and (**C**) K562 and melanoma cell line (**E**) B16F10-Nex2. Representative graph of the effect of 2′-OH-Torosaol I on the change in mitochondrial membrane potential in (**B**) Jurkat, (**D**) K562, and (**F**) B16F10-Nex2, incubated for 24 h. Results are expressed as mean ± standard error of the mean (n = 3) * *p* < 0.05 compared with the untreated control group.

**Figure 7 pharmaceuticals-16-00951-f007:**
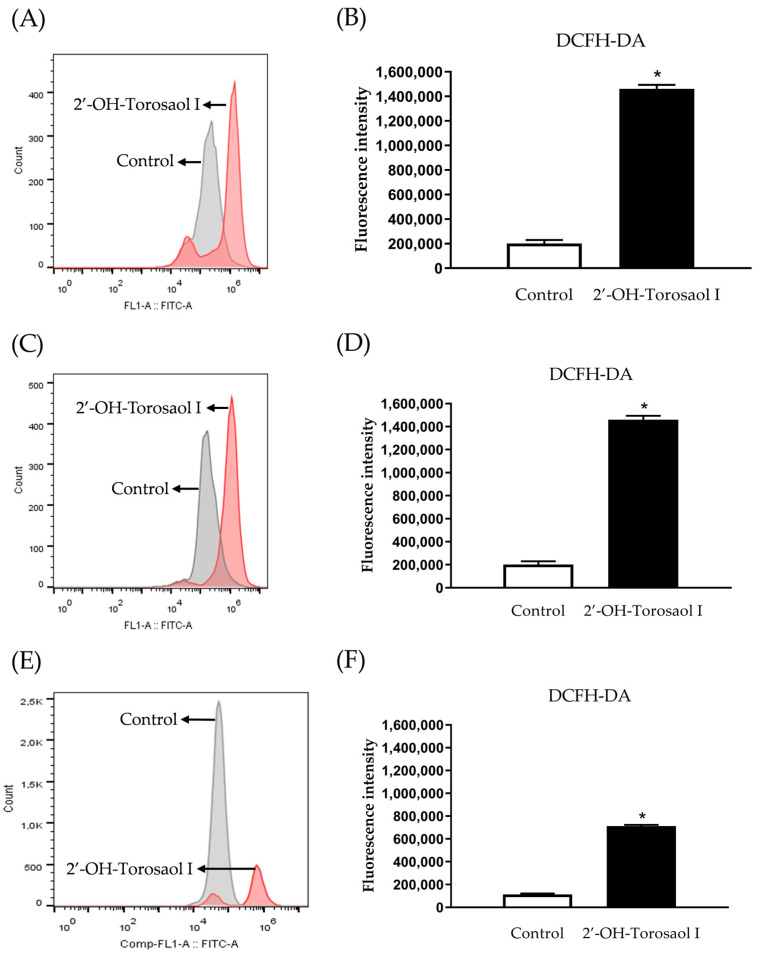
Histogram of DCFH-DA analysis by flow cytometry in leukemia cell lines (**A**) Jurkat and (**C**) K562 and melanoma cell line (**E**) B16F10-Nex2. Representative graph of the effect of 2′-OH-Torosaol I on the measurement of reactive oxygen species in (**B**) Jurkat, (**D**) K562, and (**F**) B16F10-Nex2, incubated for 24 h. Results are expressed as mean ± standard error of the mean (n = 3) * *p* < 0.05 compared with the untreated control group.

**Figure 8 pharmaceuticals-16-00951-f008:**
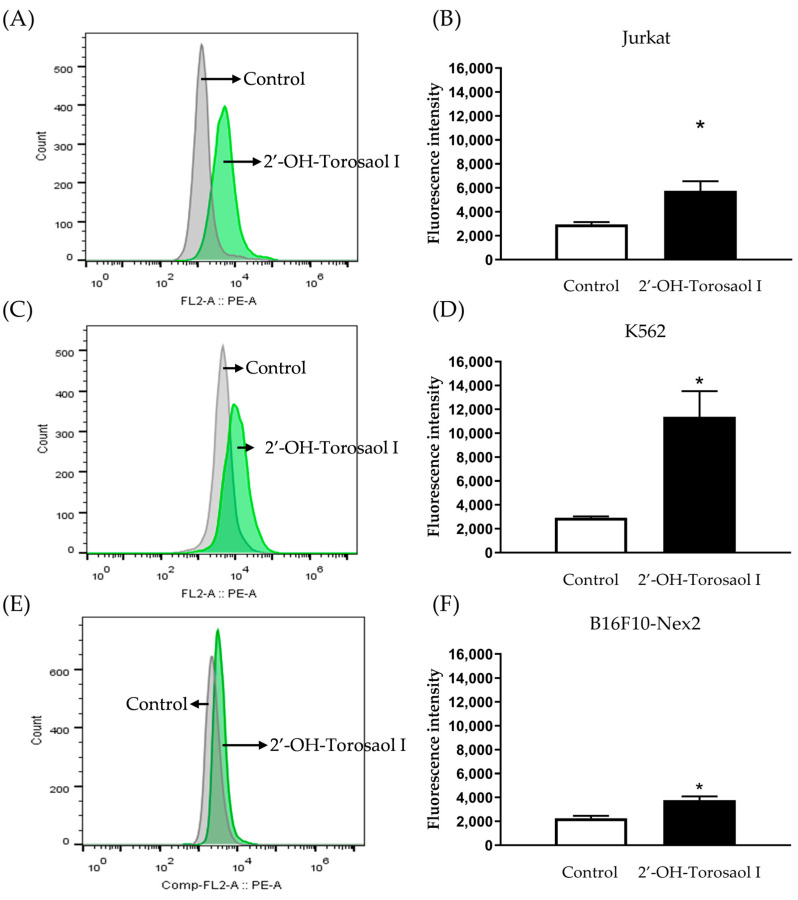
Histogram of caspase-3 analysis by flow cytometry in leukemia cell lines (**A**) Jurkat and (**C**) K562 and in melanoma cell line (**E**) B16F10-Nex2. Representative graph of the effect of 2′-OH-Torosaol I on caspase-3 activation in (**B**) Jurkat, (**D**) K562, and (**F**) B16F10-Nex2 cells, incubated for 24 h. Results are expressed as mean ± standard error of the mean (n = 3) * *p* < 0.05 compared with the untreated control group.

**Figure 9 pharmaceuticals-16-00951-f009:**
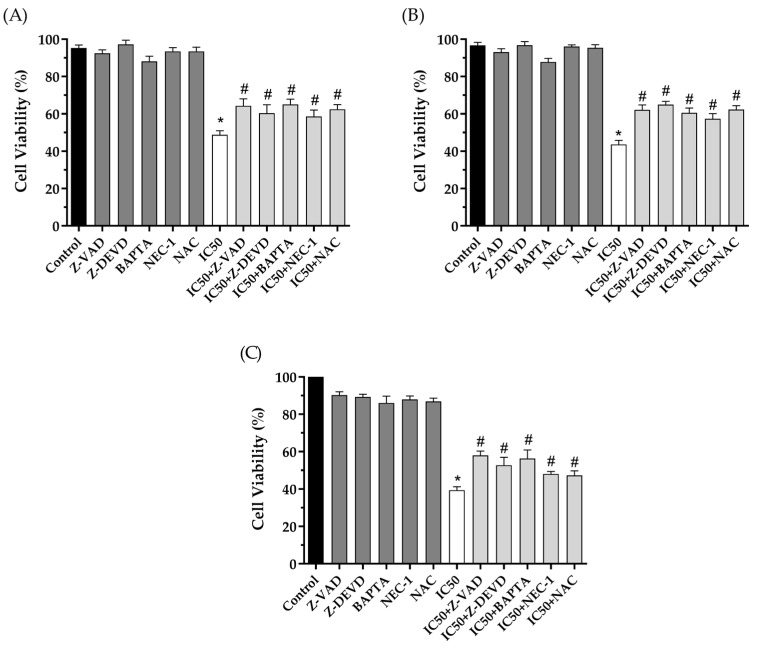
The cytotoxic effect caused by 2′-OH-Torosaol I on leukemia cell lines (**A**) Jurkat, (**B**) K562, and melanoma cell line (**C**) B16F10-Nex2, in the presence of cell death inhibitors by MTT method. Results are expressed as mean ± standard error of the mean (n = 3) * *p* < 0.05 compared with the untreated control group # *p* < 0.05 compared with the control group treated with IC_50_ of the compound.

**Table 1 pharmaceuticals-16-00951-t001:** NMR data and long-range heteronuclear correlations (DMSO-d_6_, 300 MHz) for 2′-OH-Torosaol I.

H/C	^13^C (δ)	^1^H (mult., *J* in Hz, int.)	HMBC
1,1′	205.2	-	-
2,2′	78.1	4.35 (*s*, 2H)	C-1/1′, C-3/3′, C-3/3′-Me
3,3′	73.1	-	
4,4′	39.8	2.32 (*d*, 16.8, 2H) 2.71(*d*, 16.8, 2H)	C-4/4′, C-4a/4a’, C-10/10′
4a,4a’	135.2	-	-
5,5′	95.8	5.96 (*s*, 2H)	C-7/7′-Me, C-10/10′, C-7/7′, C-10/10′
6,6′	161.2	-	-
7,7′	109.7	-	-
8,8′	155.1	-	-
8a,8a’	137.2	-	-
9,9′	162.6	-	-
9a,9a’	107.1	-	-
10,10′	122.9	-	-
10a,10a’	108.2	-	-
3,3′-Me	25.9	1.09 (*s*, 6H)	C-2/2′, C-3/3′, C-4/4′, C-4a/4a’
7,7′-Me	8.1	2.01 (*s*, 6H)	C-6/6′, C-8/8′, C-7/7′, C-8a/8a’
6,6′-OMe	55.1	3.38 (*s*, 6H)	C6/6′
3,3′-OH	-	1.17 (*s* br, 2H)	-
8,8′-OH	-	10.05 (*s* br, 2H)	-
9,9′-OH	-	15.95 (*s* br, 2H)	-

**Table 2 pharmaceuticals-16-00951-t002:** Cytotoxic potential of 2′-OH-Torosaol I on non-tumor and tumor cell lines.

	IC_50_ (µM)	
	2′-OH-Torosaol I	
Cell Lines	24 h	48 h
PBMC	> 630	> 630
B16F10-Nex2	5.8 ± 0.7	4.7 ± 0.9
SK-MEL-19	217.8 ± 12.8	139.21 ± 19.9
SK-MEL-28	108.4 ± 13.2	73.55 ± 11.0
SK-MEL-103	164.6 ± 14.6	172.5 ± 27.2
JURKAT	22.8 ± 4.5	4.89 ± 0.1
K562	21.7 ± 4.2	5.21 ± 0.8

## Data Availability

Data is contained within the article.

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
