# Peer review of "Structural Characterization and Anticancer Activity of a New Anthraquinone from Senna velutina (Fabaceae)"

_pharmaceuticals, 2023, doi:10.3390/ph16070951_

Round 1
Reviewer 1 Report
The authors David Tsuyoshi Hiramatsu Castro and co-workers submitted the manuscript entitled “Structural characterization and anticancer activity of a new anthraquinone from Senna velutina” to the journal “Pharmaceuticals” in order to be considered for publication as an “Article”.
The manuscript deals with an extract of Senna velutina (bark roots) or, more specifically, with the anthraquinone 2'-OH-torosaol I detected in the extract. The structure elucidation was based on NMR analysis. The compound was then further characterized for its anticancer activity against a range of leukemia and melanoma cell lines. In addition, it was further investigated with respect to the induction of apoptosis (flow cytometry, caspase-3), alteration of mitochondrial membrane potential and generation of ROS. To further narrow down the mechanism of cell death, experiments with specific cell death inhibitors were performed. The subject matter of the publication fits with the special issue "Pharmacology and Medicinal Value of Flowering Plant in the Treatment of Non-communicable Diseases." Basically, the authors present a comprehensive study, although some aspects would help to further optimize the manuscript:
@ “From the extract chloroform of root barks of S. velutina, the new substance 1 was isolated and revealed a chromatographic purity of 93% by LC-MS.” That may well be the case, but an extract is usually made up of several components. In order to provide a clear presentation, the authors must include the following aspects in their manuscript: on the one hand, what type of chloroform extract is it in particular? More detailed information is needed. On the other hand, a profiling of the composition of the extract, e.g. by GC-MS analysis, should be done. From this it will probably be deduced that "substance 1" is the most abundant and therefore considered as one of the effect determining components... and consequently this compound was considered further. As it is still the case that "substance 1" is simply presented to the readers, this is not sufficiently comprehensible.
As far as my understanding goes, 2'-OH-torosaol I is equivalent to 2,2'-OH-singueanol I. The authors should mention this in their manuscript because otherwise it could lead to confusion for the reader.
Table 2: The representation of the values is not suitable. First, there is the question of the number of repetitions (n=?). And then it cannot be that the SEM is given with more decimal places (thus supposedly more exact) than the mean value. All in all, a consideration of the uniformity concerning valid digits can also be considered.
Why was the cytotoxicity not determined in the non-cancer cell line PBMC? Why is it mentioned in the manuscript that it was done? This does not fit. However, when assessing the general toxicity of potential anticancer compounds, a study with non-cancer cell lines is highly recommended. The authors are requested to submit these experiments subsequently. In this case, the IC50 value is >630 micromolar.
The authors have examined the cytotoxic profile after two different time points, but there is no discussion of this in section 2.2. Therefore, the authors are kindly asked to provide this.
Although it is mentioned in the body text that a comparison with a control is made for the biological effects, it is not further specified what exactly this control is. Positive or negative control, which substance or even without?
Why are the figures in Figures 5 A/C/E not also colored as in Figures 6 A/C/E? That would be more reader-friendly.
@ line 256 “DCF-DA“, please introduce abbreviation. This is generally true. Before the continuous use of abbreviations, they must be explained the first time they are mentioned.
The authors used different inhibitors to further characterize the modality of cell death. This is a great approach! Since there was a significant increase of ROS, besides the investigation of necroptosis by NEC-1, an investigation of the newly discovered mechanism ferroptosis by FER-1 (see for example https://doi.org/10.1039/D0DT00168F) is recommended. Ferroptosis is not limited to iron-containing compounds, but is characterized by an increase in lipid-based ROS. It is a non-apoptotic mechanism and the extent of induction of caspase-3 apparently does not cover the extent of ROS formation, suggesting just another cell death pathway. The addition of FER-1 to the experiment is therefore highly recommended.
Line 410: Ca+ vs. Ca2+ ?
@ “…and identified by a botanist…” On which parameters was the characterization based? Please note accordingly in the manuscript.
@ “and 12x204 for PBMC in 96-well…” Is this really right? Please check.
Although the paper is edited by the MDPI publisher, the authors are asked to correct a number of formal aspects. The following list is not intended to be exhaustive, but to assist the authors in addressing these issues:
Lines 13/Table 1/paragraph 2: inconsistency regarding font; Line 43: presentation of the keywords; Lines 52/69/87/89/315/316/317/369/377: remove redundant brackets; Line 73: introduce abbreviation of S. velutina before consistently using the abbreviation throughout the manuscript; Line 74: case shift (also 208 vs. 234 vs. 525 vs. 267, also 567 vs. 568); Lines 72/74/78/83/84: avoid repetition of demonstrate by using different vocabulary (also in the paragraph “Discussion”); Line 92: remove comma; paragraph 2: line distance; Line 127: formatting; line 130/163/174/205/451/452/473/493/507/509/510/518/519/524/535/560/561/570/572/573/582/584/585: use of space with units (also Figures 2, 3, 4); Lines 162/187: and; Line 169: symbol; Lines 168 vs. 171: symbols; Line 172/233/484/493/518/534/535/550/559/583: subscript; Line 324: punctuation; Lines 371/374: presentation of citations; Line 481/482/483/496: superscript; Line 512: use space in the formula for better readability; Line 520: symbol micro insead of u; Line 580/582: in italics; among others.
In my eyes, moderate English editing is recommended, because I read quite a few parts of the manuscript a bit bumpy. Well, English is spoken a little differently in different parts of the world, but the meaning of what is said is not affected. Nevertheless, in my opinion, it sometimes does not correspond to the usual scientific English.
Author Response
@ “From the extract chloroform of root barks of S. velutina, the new substance 1 was isolated and revealed a chromatographic purity of 93% by LC-MS.” That may well be the case, but an extract is usually made up of several components. In order to provide a clear presentation, the authors must include the following aspects in their manuscript: on the one hand, what type of chloroform extract is it in particular? More detailed information is needed. On the other hand, a profiling of the composition of the extract, e.g. by GC-MS analysis, should be done. From this it will probably be deduced that "substance 1" is the most abundant and therefore considered as one of the effect determining components... and consequently this compound was considered further. As it is still the case that "substance 1" is simply presented to the readers, this is not sufficiently comprehensible.
Answer: The quoted text in the results has been revised to provide more precise information regarding the isolation of the compound using classical techniques, specifically fractionation column chromatography. The compound was subsequently characterized using mass spectrometry and NMR. It is important to note that the detailed procedure for isolating the compound can be found in section 4.2 of the materials and methods. This section elaborates on the source of the chloroform extract and outlines all the essential steps required for methodological reproducibility. This information is handy for readers interested in obtaining the compound.
“4.2. The roots extract fractionating process
- velutina roots (395 g) were extracted by exhaustive percolation ethanol-water (7:3), for 48h, dripping 45 drops per minute to obtain the ethanolethanolic extract. The collected material was rotary evaporated at 40 °C40°C. The extract was submitted to liquid-liquid extraction with chloroform to obtain the chloroform extract (15.5 g). This extract was fractionated in classic silica gel column chromatography (Mesh 5 to 8) in elution gradient with chloroform and hexane 6:4 (v/v), chloroform and hexane 8:2 (v/v), chloroform, chloroform and methanol 1%1 %, chloroform and methanol 3%3 %, chloroform and methanol 10%10 % and methanol. In the end, 30 fractions were obtained. Fractions 23-26 showed a similar chemical profile and were put together, obtaining 450 mg of a compound that was analyzed by HPLC-DAD-MS and structurally characterized by NMR as 2'-OH-Torosaol I, which is also referred to as 2,2'-OH-singueanol I (93% purity grade), a novel substance in the literature.”
As far as my understanding goes, 2'-OH-torosaol I is equivalent to 2,2'-OH-singueanol I. The authors should mention this in their manuscript because otherwise it could lead to confusion for the reader.
Answer: Thank you for your valuable feedback. We agree that it's important to explicitly state the correlation between 2'-OH-torosaol I and 2,2'-OH-singueanol I in our paper. We have included both names in the updated manuscript.
Table 2: The representation of the values is not suitable. First, there is the question of the number of repetitions (n=?). And then it cannot be that the SEM is given with more decimal places (thus supposedly more exact) than the mean value. All in all, a consideration of the uniformity concerning valid digits can also be considered.
Answer: Thanks for this observation, we have corrected the variations in the decimal places as well as included in the figure legend description of the number of replicates of the experiments.
Why was the cytotoxicity not determined in the non-cancer cell line PBMC? Why is it mentioned in the manuscript that it was done? This does not fit. However, when assessing the general toxicity of potential anticancer compounds, a study with non-cancer cell lines is highly recommended. The authors are requested to submit these experiments subsequently. In this case, the IC50 value is >630 micromolar.
Answer: Our sincere apologies for our mistake, we would like to thank reviewer 1 for this observation. Now in the new version of the manuscript, the IC50 is correctly represented above 630 µM/mL in the case of PBMC.
The authors have examined the cytotoxic profile after two different time points, but there is no discussion of this in section 2.2. Therefore, the authors are kindly asked to provide this.
Answer: Thanks for this important observation. We have revised the presentation of results in section 2.2 to reflect these results.
Although it is mentioned in the body text that a comparison with a control is made for the biological effects, it is not further specified what exactly this control is. Positive or negative control, which substance or even without?
Answer: We have included a statement in the method section 4.6 of what the control group represents. In all cases, the control refers to untreated cells representing 100% viability.
Why are the figures in Figures 5 A/C/E not also colored as in Figures 6 A/C/E? That would be more reader-friendly.
Answer: Following the suggestion of the reviewer, the figures in Figures 5 A/C/E also are colored.
@ line 256 “DCF-DA“, please introduce abbreviation. This is generally true. Before the continuous use of abbreviations, they must be explained the first time they are mentioned.
Answer: We have revised this in the manuscript. In fact “DCF-DA” is a typo, it should be DCFH-DA. This error has been corrected in the manuscript.
The authors used different inhibitors to further characterize the modality of cell death. This is a great approach! Since there was a significant increase of ROS, besides the investigation of necroptosis by NEC-1, an investigation of the newly discovered mechanism ferroptosis by FER-1 (see for example ttps://doi.org/10.1039/D0DT00168F) is recommended. Ferroptosis is not limited to iron-containing compounds, but is characterized by an increase in lipid-based ROS. It is a non-apoptotic mechanism and the extent of induction of caspase-3 apparently does not cover the extent of ROS formation, suggesting just another cell death pathway. The addition of FER-1 to the experiment is therefore highly recommended.
Answer: Thanks for this important suggestion. We are aware of the different mechanisms that may be explored to characterize the death pathways, however, considering the principal objectives of this present study were “to isolate, identify, characterize the chemical structure of the major anthraquinone identified in the extract of S. velutina, roots and in addition, to evaluate its cytotoxic potential against melanoma and leukemic cell lines and identified possible death pathways involved”, which we believe have been accomplished with the present results. It is part of our future studies to further explore the mechanisms involved in the compound anti-tumoral activities. Among the mechanisms to be studied in the future is ferroptosis by FER-1. In addition to the points raised, it is worth mentioning that there are logistical and time limitations for carrying out experiments quickly. This could also negatively impact the novelty of the new molecule presented in this study.
Line 410: Ca+ vs. Ca2+ ?
Answer: This point was revised and changed in the manuscript.
@ “…and identified by a botanist…” On which parameters was the characterization based? Please note accordingly in the manuscript.
Answer: We would like to thank the reviewer for this observation. In the new version of the manuscript, in section 4.1 of the materials and methods, the parameters applied for the characterization of the botanic of Senna velutina were included.
@ “and 12x204 for PBMC in 96-well…” Is this really right? Please check.
Answer: Thanks for the observation, it was a typographical error. It should read as 12x104 “ This has been corrected in the manuscript.
Although the paper is edited by the MDPI publisher, the authors are asked to correct a number of formal aspects. The following list is not intended to be exhaustive, but to assist the authors in addressing these issues:
Lines 13/Table 1/paragraph 2: inconsistency regarding font;
Answer: Corrected as noted.
Line 43: presentation of the keywords;
Answer: Corrected
Lines 52/69/87/89/315/316/317/369/377: remove redundant brackets;
Answer: Corrected in the manuscript.
Line 73: introduce abbreviation of S. velutina before consistently using the abbreviation throughout the manuscript;
Answer: Corrected as noted.
Line 74: case shift (also 208 vs. 234 vs. 525 vs. 267, also 567 vs. 568);
Answer: Corrected as noted.
Lines 72/74/78/83/84: avoid repetition of demonstrate by using different vocabulary (also in the paragraph “Discussion”);
Answer: Corrected as suggested in the revised manuscript.
Line 92: remove comma; paragraph 2: line distance; Line 127: formatting; line 130/163/174/205/451/452/473/493/507/509/510/518/519/524/535/560/561/570/572/573/582/584/585:
Answer: Corrected as suggested
use of space with units (also Figures 2, 3, 4); Lines 162/187: and; Line 169: symbol; Lines 168 vs. 171: symbols; Line 172/233/484/493/518/534/535/550/559/583: subscript;
Line 324: punctuation; Lines 371/374:
Answer: All corrections made accordingly
presentation of citations; Line 481/482/483/496: superscript; Line 512: use space in the formula for better readability; Line 520: symbol micro insead of u; Line 580/582: in italics; among others.
Answer: All corrections made accordingly
Comments on the Quality of English Language
In my eyes, moderate English editing is recommended, because I read quite a few parts of the manuscript a bit bumpy. Well, English is spoken a little differently in different parts of the world, but the meaning of what is said is not affected. Nevertheless, in my opinion, it sometimes does not correspond to the usual scientific English.
Answer: We have revised the English language and made the necessary corrections. One of us who is a native English speaker with several years of experience in writing and publishing in life sciences journals did the translation and editing of the final revised manuscript.

Reviewer 2 Report
Manuscript was written well and characterize the major metabolites and their anticancer activity. The research looks promising though the minor issues needs to be resolved before publication.
Title: “: Structural characterization and anticancer activity of a new anthraquinone from Senna velutina” could be “: Structural characterization and anticancer activity of a new anthraquinone from Senna velutina (Fabaceae)”.
In Abstract: Line (14-17) looks very general can be moved to introduction or can be removed. Abstract needs to be start with the research gaps and the outcomes.
Line 23: “barks of the roots of” should be “root barks of”.
Expand NMR and MTT in abstract since it is used for the first time.
In Abstract Keywords: “keyword” is that a keyword? can be changed to other catchy words which is relevant to the research.
Line 47-48: “Among the types of cancers are leukemias and melanoma” needs to be rewritten looks lucid.
Line 52: “according to the progression” should be “based on the progression”. Also, ([2] should be “[2]”. Similarly, Line 69: ([12,13] should be “[12,13]
Line 73: “S. velutina” should be “Senna velutina” use whole genus name since it is mentioned for first time.
Line 74: “CASTRO” should be “Castro”. Also, “we demonstrated the” should be “demonstrated that”.
Line 75: “in vitro and in vivo models” needs to be italics.
Line 81-83: The sentence “Of the identified constituents, anthraquinones have great prominence as……..” its looks fragmented needs to be rewritten.
Kindly check the cited reference style and format as per the journal standards most of them are cited wrong as “([18-19]. Check the entire manuscript and revise.
Line 91: “and determine” can be removed.
Line 92: “identified in the extract of” should be “from the”. Similarly “In addition, to evaluate its cytotoxic potential” Also, its cytotoxic potential has been evaluated”.
Line 103-104: “which are compatible with bis-tetrahydro-anthracene chromophore similar to those isolated of roots from Senna singueana” How come reference has been cited in the results part it needs to be moved to discussion part.
Legends of Figure 1 has been missing.
In Table 1: What is the symbol “-“ denotes. Need explanation in the table legends.
There is no sub-heading for the section 2.2.
Table 2. Kindly include the “Results were expressed as mean ± standard error of the mean (n=3)” in the table legends.
Figure. 2 Why all Y axis scale were up to 120% as it is measured as 100%?.
Line 166: Same here “both in vitro and in vivo” should be italics.
Figure 3: All figures needs to be labelled with their magnification range, which is captured, For eg:- “-20µm-” or “-50µm-“ label as per the range. Also, all images looks similar can you highlight by marking any changes between the treatment hours (0h to 48h).
Figure. 4. As similar to fig.2 “Why all Y axis scale were up to 120% as it is measured 100%?
Figure. 5 A, C, and E images X and Y axis looks small can be enlarged and improve the resolution for better view. As similar in Fig. 6 A, C and E.
Conclusion seems to be lucid needs to be added with the present novel outcome and research gap followed by the future perspective of the research.
Line 540: “evaluated in FlowJo 10.6” need clarification of this include make and model if it is an instrument.
Minor Grammar and Typo errors need to be corrected.
Author Response
We would like to thank once again reviewer 2 for their careful evaluation and suggestions related to our manuscript.
Manuscript was written well and characterize the major metabolites and their anticancer activity. The research looks promising though the minor issues needs to be resolved before publication.
Title: “: Structural characterization and anticancer activity of a new anthraquinone from Senna velutina” could be “: Structural characterization and anticancer activity of a new anthraquinone from Senna velutina (Fabaceae)”.
Answer: We believe the suggestion is germane and has therefore altered the title accordingly.
In Abstract: Line (14-17) looks very general can be moved to introduction or can be removed. Abstract needs to be start with the research gaps and the outcomes.
Answer: Corrected as suggested.
Line 23: “barks of the roots of” should be “root barks of”.
Answer: Altered as suggested.
Expand NMR and MTT in abstract since it is used for the first time.
Answer: Corrected as suggested
In Abstract Keywords: “keyword” is that a keyword? can be changed to other catchy words which is relevant to the research.
Answer: Thanks for calling our attention to this error. It was a typo and has been corrected.
Line 47-48: “Among the types of cancers are leukemias and melanoma” needs to be rewritten looks lucid.
Answer: Corrected as suggested
Line 52: “according to the progression” should be “based on the progression”. Also, ([2] should be “[2]”. Similarly, Line 69: ([12,13] should be “[12,13]
Answer: Corrected as advised.
Line 73: “S. velutina” should be “Senna velutina” use whole genus name since it is mentioned for first time.
Answer: Corrected as mentioned.
Line 74: “CASTRO” should be “Castro”. Also, “we demonstrated the” should be “demonstrated that”.
Answer: Corrected as suggested.
Line 75: “in vitro and in vivo models” needs to be italics.
Answer: This point was revised and changed in the manuscript.
Line 81-83: The sentence “Of the identified constituents, anthraquinones have great prominence as……..” its looks fragmented needs to be rewritten.
Answer: Corrected as suggested.
Kindly check the cited reference style and format as per the journal standards most of them are cited wrong as “([18-19]. Check the entire manuscript and revise.
Answer: All typos have been corrected in the manuscript.
Line 91: “and determine” can be removed.
Answer: Corrected
Line 92: “identified in the extract of” should be “from the”. Similarly “In addition, to evaluate its cytotoxic potential” Also, its cytotoxic potential has been evaluated”.
Answer: All corrected in the manuscript.
Line 103-104: “which are compatible with bis-tetrahydro-anthracene chromophore similar to those isolated of roots from Senna singueana” How come reference has been cited in the results part it needs to be moved to discussion part.
Answer: We have excluded this part from the section. We agreed it should only be in the discussion part.
Legends of Figure 1 has been missing.
Answer: Thanks for the observation. We have added the missing legends and made necessary corrections to others as well.
In Table 1: What is the symbol “-“ denotes. Need explanation in the table legends.
Answer: Deleted from the figure.
There is no sub-heading for the section 2.2.
Answer: Thanks for the observation. The sub-heading has now been included “Effect of 2'-OH-Torosaol on Cell viability”
Table 2. Kindly include the “Results were expressed as mean ± standard error of the mean (n=3)” in the table legends.
Answer: Corrected as suggested.
Figure. 2 Why all Y axis scale were up to 120% as it is measured as 100%?.
Answer: Thanks for the observation, we have corrected for this in the revised manuscript.
Line 166: Same here “both in vitro and in vivo” should be italics.
Answer: This point was revised and changed in the manuscript.
Figure 3: All figures needs to be labelled with their magnification range, which is captured, For eg:- “-20µm-” or “-50µm-“ label as per the range. Also, all images looks similar can you highlight by marking any changes between the treatment hours (0h to 48h).
Answer: Thanks for the observations, we have provided better quality images for these figures in the revised manuscript, and included the scale.
Figure. 4. As similar to fig.2 “Why all Y axis scale were up to 120% as it is measured
Answer: Thanks for the observations, we have revised the figures accordingly.
Figure. 5 A, C, and E images X and Y axis looks small can be enlarged and improve the resolution for better view. As similar in Fig. 6 A, C and E.
Answer: Thanks for the observations, we have provided better quality images for these figures in the revised manuscript.
Conclusion seems to be lucid needs to be added with the present novel outcome and research gap followed by the future perspective of the research.
Answer: Thanks for the suggestion, we have updated the concluding part of the manuscript to reflect this.
Line 540: “evaluated in FlowJo 10.6” need clarification of this include make and model if it is an instrument.
Answer: Flowjo is a flow cytometer software for analyzing images.
We have checked all other errors and made general revision of the text for clarity, language and aesthetics changes. We here submit two versions of the revised manuscript (clean and marked).
Round 2
Reviewer 1 Report
The authors David Tsuyoshi Hiramatsu Castro and co-workers submitted a revised version of their manuscript “Structural characterization and anticancer activity of a new anthraquinone from Senna velutina (Fabaceae)” to the journal “Pharmaceuticals”.
The authors have referred to the information in subsection 4.2 for the characterization of the extraction process. Nevertheless, it is not clear how the individual fractions were characterized. It is only said that "Fractions 23-26 showed a similar chemical profile and were put together", but what this means exactly is unclear. How does one come to take exactly these fractions 23-26. Only afterwards the analytical characterization of these fractions was done. What about the other fractions? I still find this aspect unclear.
The authors did not perform the proposed experiments with the specific ferroptosis inhibitor FER-1. They justify this by the fact that even without these experiments the objectives of the publication ("to isolate, identify, characterize the chemical structure of the major anthraquinone identified in the extract of S. velutina, roots and in addition, to evaluate its cytotoxic potential against melanoma and leukemic cell lines and identified possible death pathways involved") have already been worked off. The study on the "identification of possible death pathways involved" would complement this exactly. In addition, the authors state that it would be difficult to implement this in terms of time. After transparent consultation with the editor, an extension of the time for the revision would also be possible. It is a single substance and the authors would not have to show this on all cancer cell lines. For example, the most active cell lines would be sufficient. I think this aspect can be included, as it should not take so much time.
Otherwise, the authors have optimized the illustrations where necessary, explained abbreviations, corrected typos, provided explanations in the manuscript etc. Formal errors have been largely corrected. The remaining minor issues will be fixed in the course of editing or proof-reading.
All the best!
Minor editing of English suggested.
Author Response
The authors David Tsuyoshi Hiramatsu Castro and co-workers submitted a revised version of their manuscript “Structural characterization and anticancer activity of a new anthraquinone from Senna velutina (Fabaceae)” to the journal “Pharmaceuticals”.
The authors have referred to the information in subsection 4.2 for the characterization of the extraction process. Nevertheless, it is not clear how the individual fractions were characterized. It is only said that "Fractions 23-26 showed a similar chemical profile and were put together", but what this means exactly is unclear. How does one come to take exactly these fractions 23-26. Only afterwards the analytical characterization of these fractions was done. What about the other fractions? I still find this aspect unclear.
We would like to thank reviewer 1 for the useful comments, criticism, and suggestions. It has helped in improving the quality of the manuscript. The details of what exactly was done have now been included in the manuscript. It now reads as follows:
In all, 30 fractions were obtained. Based on a thorough analysis using HPLC-DAD-MS, it became clear that the fractions 23-26 had almost identical chemical profiles. As a result, these fractions were combined for further examination., obtaining 450 mg that was reanalyzed by HPLC-DAD-MS and structurally characterized by NMR as 2'-OH-Torosaol I, which is also referred to as 2,2'-OH-singueanol I (93% purity grade), a novel substance in the literature.
The authors did not perform the proposed experiments with the specific ferroptosis inhibitor FER-1. They justify this by the fact that even without these experiments the objectives of the publication ("to isolate, identify, characterize the chemical structure of the major anthraquinone identified in the extract of S. velutina, roots and in addition, to evaluate its cytotoxic potential against melanoma and leukemic cell lines and identified possible death pathways involved") have already been worked off. The study on the "identification of possible death pathways involved" would complement this exactly. In addition, the authors state that it would be difficult to implement this in terms of time. After transparent consultation with the editor, an extension of the time for the revision would also be possible. It is a single substance and the authors would not have to show this on all cancer cell lines. For example, the most active cell lines would be sufficient. I think this aspect can be included, as it should not take so much time.
We indeed consider this suggestion to be germane, however, reiterate that the manuscript presents information on mechanisms of death induced by the new molecule described. Furthermore, in future studies, not only ferroptosis, but also necroptosis, oncosis, pyroptosis, parthanatos, and autophagic cell death will be evaluated as potentially involved. We reaffirm that our current possibility of carrying out the experiment suggested as the use of the inhibitor is limited by the fact that the inhibitor to be used for this assay is not readily available in the country (Brazil). In order to import, we would need to apply for direct importation as a researcher, for which a minimum of 90 days is expected for a definite response from the regulatory agency. We would like to meet the period given by the Pharmaceuticals Journal of 10 days for these corrections to be made, which in our opinion does not harm the manuscript. Cordially, we hope that reviewer 1 understands this logistical problem that we are subject to in Brazil, without prejudice to the publication of this study.
Round 3
Reviewer 1 Report
The authors David Tsuyoshi Hiramatsu Castro and colleagues once more provided a revised version of their manuscript “Structural characterization and anticancer activity of a new anthraquinone from Senna velutina (Fabaceae)” submitted to the journal “Pharmaceuticals”.
In their revision, the authors provided additional explanation and added the appropriate addendum to their manuscript as to why they focused on fractions 23-26 of the extraction. This explanation is reasonable as far as it goes.
Regarding the requested experiment with the ferroptosis inhibitor FER-1, the authors maintain that they will not perform the experiments. They repeatedly state that they want to work on this in a future study. Furthermore, they state that they have severe difficulties in acquiring the substance FER-1.
minor editing suggested